# Wear Resistance of Light-Cure Resin Luting Cements for Ceramic Veneers

**DOI:** 10.3390/jfb16010005

**Published:** 2024-12-26

**Authors:** Miyuki Oshika, Takafumi Kishimoto, Taku Horie, Abdulaziz Alhotan, Masao Irie, Veronica C. Sule, Wayne W. Barkmeier, Akimasa Tsujimoto

**Affiliations:** 1Department of Operative Dentistry, School of Dentistry, Aichi Gakuin University, Nagoya 464-8651, Japan; oshika@dpc.agu.ac.jp (M.O.); taka-ki@dpc.agu.ac.jp (T.K.); lifedoor@dpc.agu.ac.jp (T.H.); 2Department of Oral Health Sciences, BIOMAT, KU Leuven, 3000 Leuven, Belgium; 3Department of Dental Health, College of Applied Medical Sciences, King Saud University, P.O. Box 10219, Riyadh 12372, Saudi Arabia; aalhotan@ksu.edu.sa; 4Department of Biomaterials, Okayama University Graduate School of Medicine, Dentistry and Pharmaceutical Sciences, 2-5-1 Shikata-cho, Kita-ku, Okayama 700-8525, Japan; mirie@md.okayama-u.ac.jp; 5Department of General Dentistry, Creighton University School of Dentistry, Omaha, NE 68102, USA; veronicasule@creighton.edu (V.C.S.); waynebarkmeier@creighton.edu (W.W.B.); 6Department of Operative Dentistry, University of Iowa College of Dentistry, Iowa City, IA 52242, USA

**Keywords:** dental biomaterials, surface analysis, wear, cementation, veneer

## Abstract

The purpose of this study was to compare the wear resistance of light-cure resin luting cements for veneers with that of other luting materials investigated in earlier studies. An Alabama wear-testing machine was used to measure the wear resistance of four recent light-cure resin luting cements for veneers (G-Cem Veneer; Panavia V5 LC; RelyX Veneer Cement; and Vario-link Esthetic LC). The volume loss ranged from 0.027 ± 0.003 to 0.119 ± 0.030 mm^3^, the mean facet depth from 56.053 ± 7.074 to 81.531 ± 7.712 µm, and the maximum facet depth from 100.439 ± 26.534 to 215.958 ± 27.320 µm. G-Cem Veneer showed significantly better (*p* < 0.05) wear resistance than the other materials tested. Representative SEM images were obtained which showed differences in form among the wear facets for the luting cements examined. Correlations were calculated between the three measurements for each material, and the pattern of correlations was also different for each material.

## 1. Introduction

Ceramic veneers are indirect aesthetic restorations with minimal tooth reduction, used to enhance the color, form, and function of unaesthetic teeth [1]. Light-cure resin luting cements are used for the bonding of veneers because the composition of the cements has better color stability than dual-cure resin luting cements [2]. These light-cure resin luting cements are used with primers, and therefore acidic functional monomers are not included in the cement itself. This is thought to contribute to the chemical stability of the material over the long term, and to play an important role in the quality of the bond between the veneer and the underlying tooth substrate. There is a long history and there have been long-term clinical studies on the use of ceramic veneers, and those studies showed excellent clinical performance of ceramic veneers over periods of 20 years, with more than 80% clinical survival rates [3,4].

However, one of the problems of long-term usage of veneers (3 surfaces) is gap formation between the veneer and the surrounding tooth structure, as shown in Figure 1, which shows a veneer after 18 years of clinical service. Although the veneer is still sound and shows good adaptation, the margin on the lingual side shows a clear gap between the veneer and the tooth substrate. This gap formation is mainly due to wear of the light-cure resin luting cement. This sort of gap is an ideal site for the retention of plaque, is vulnerable to staining, and may serve as a starting point of failure [5]. Also, if the worn surface of the cement becomes rougher, the wear rate will be accelerated by the abrasiveness of the surface [6]. These problems reduce the quality of the restoration even if they do not lead to failure, and therefore it is preferable to use a cement that minimizes them.

Recent developments in ceramics have made this problem even more pressing. Stronger ceramics, such as leucite and lithium disilicate reinforced ceramics, and translucent zirconia, have enabled the use of ultra-thin, thin and partial ceramic veneers with minimal preparations [7], and the creation of occlusal veneers, which do not extend all the way down the tooth [8]. With conventional crowns, the margin of the crown was mainly located in the gingival or subgingival area, and thus did not create much of an aesthetic or wear problem [9]. However, with anterior (three surfaces) or occlusal veneers, the cement line is visible on the outer surface of the tooth, and thus wear at that location may create aesthetic problems in addition to clinical issues. Furthermore, the joint line may be more exposed to wear during mastication, increasing the degree of wear beyond that seen with conventional crowns. Wear of the cement line on such restorations may also create an irregularity on the exposed surface of the tooth, which could in turn accelerate the wear of the antagonist surface.

This has made clinicians more concerned about the wear resistance of resin luting cements used to secure the ceramic veneer to tooth structures. Generally, the use of light-cure resin luting cements has been recommended for partial coverage restorations in order to avoid this kind of problem [2]. Although dual-cure resin luting cements are similar in many ways to light-cure versions, they are vulnerable to discoloration over the medium to long term, as most of the materials contain amines for the chemical activation of polymerization. Therefore, they were not preferred options for use in partial coverage aesthetic restorations [10]. The wear resistance of restorative resin-based composites has improved substantially in recent years [11]. Flowable resin-based composites (RBCs) have also been used [12], but similarly face several problems. First, their viscosity is still often higher than that of cement, particularly for injectable RBCs, which makes it difficult to obtain good adaptations and fill-thickness. Second, it has been generally considered that their wear resistance is still inferior to that of paste-type RBCs [13]. Third, the use of flowable RBCs for ceramic restorations generally provides less reliable bonding than that of a dual-cure resin luting cement [14]. Some clinicians have therefore tried using paste-type RBCs, but in this case it is necessary to heat the material to around 54 °C in order to reduce the viscosity to a point where it can be applied [15]. This heating has unknown effects on the properties of the RBC itself, and is also high enough to potentially have adverse effects on the pulp through the thin layer of enamel and dentin remaining. Furthermore, RBCs are not cements, and thus not ideally suited to this role.

These problems could be avoided if there were light-cured resin luting cements with similar wear resistance to the other classes of material. In the past, the Alabama wear-testing machine has been used to measure the wear resistance of materials in all these classes [16,17,18]. Using the same methodology ensures the comparability of results for different materials, but there are no reported studies using this technique to investigate the wear resistance of recently introduced light-cured resin luting cements.

Therefore, the purpose of this study is to evaluate the wear resistance of recently introduced light-cure resin luting cements through measurements of volume loss and both mean and maximum depth after simulated wear in an Alabama wear-testing machine, and compare with the alternative materials to determine whether light-cure resin luting cements are a reasonable clinical choice.

## 2. Materials and Methods

### 2.1. Study Materials

Four veneer cements: (1) G-Cem Veneer (GV, GC, Tokyo, Japan); (2) Panavia V5 LC (PV, Kuraray Noritake Dental, Tokyo, Japan); (3) RelyX Veneer Cement (RV, 3M ESPE, Seefeld, Germany); and (4) Variolink Esthetic LC (VE, Ivocla Vivadent, Schaan, Liechtenstein), were used in this laboratory investigation (Table 1).

### 2.2. Specimen Preparation

Previous research using an Alabama wear testing machine (University of Alabama at Birmingham, Birmingham, AL, USA) used a range of sample sizes, from 6 to 20 specimens in each group [16,17,18] (Figure 2). The sample size was determined based on effect size = 0.25 (medium) or 0.5 (large), a = 0.05, power = 0.8, and number of groups = 4. The result was that 180 specimens (medium effect size) or 48 (large effect size) were needed. With four groups, either 12 or 45 specimens were needed in each group, and 12 was selected.

The samples were prepared following the methods described by Tsujimoto et al. [16]. Polishing, as described in reference [16], was conducted in order to approximate the clinical situation, in which these materials are polished.

### 2.3. Wear Testing

An Alabama wear-testing machine was used in this study [16,17,18], and the protocol described by Tsujimoto et al. [16] was followed.

### 2.4. Measurement of Wear Facet

The wear facets were measured as described in Tsujimoto et al. [16].

### 2.5. Statistical Analysis

A commercial statistical software package (IBM SPSS version 29.0.2.0, Chicago, IL, USA) was used. The Shapiro–Wilk test was used to confirm that the data were normally distributed. A one-way ANOVA (analysis of variance) with Tukey’s post hoc honest significant difference test was used. VL MD and AD were analyzed with a significance level of 0.05. Additionally, Pearson correlation analysis was performed for all materials as well as between the VL, MD, and AD for each material.

### 2.6. Scanning Election Microscopy (SEM) Observations

Samples were chosen randomly for SEM observation using a Tabletop Microscopes (TM3000, Hitachi, Tokyo, Japan). A thin coating of an 80/20 gold–palladium alloy was applied in a sputter coater (Emitech SC7620 Mini Sputter Coaterk, Quorum Technologies, East Sussex, UK) to prevent accumulation of electrostatic charge at the sample surface. An operating voltage of 15 kV and magnifications of 100× and 2500× were used.

## 3. Results

### 3.1. Localized Wear

The results for localized wear (VL, MD and AD (Mean (average) depth)) are presented in Table 2. The one-way ANOVA among VL, MD and AD showed a significant difference (*p* < 0.05) for the material factor. Tukey’s post hoc test for VL, MD and AD showed significant differences (*p* < 0.05) in localized wear among the materials tested. The VLs of the materials evaluated in this study ranged from 0.027 ± 0.003 to 0.119 ± 0.030 mm^3^. The VL of GV was significantly lower (*p* < 0.05) than that of the other three materials evaluated in this study. No significant difference (*p* > 0.05) in VL was found between VE and either PV or RV, although PV and RV were significantly different (*p* < 0.05) from each other. The MDs for materials in this study ranged from 100.439 ± 26.534 to 215.958 ± 27.320 µm. The MD of GV was significantly less (*p* < 0.05) than that of the other three materials evaluated in this study. The MD of PV was significantly greater (*p* < 0.05) than that of the other three materials evaluated in this study. The ADs for materials in this study ranged from 56.053 ± 7.074 to 81.531 ± 7.712 µm. The AD of GC was significantly less (*p* < 0.05) than that of VE, but there were no significant differences between the other materials.

### 3.2. Correlation Analysis

The mean VL of all materials was 0.078 ± 0.039 mm^3^, the mean MD of all materials was 153.034 ± 47.744 µm and the mean AD of all materials was 70.182 ± 15.371 µm. VL and MD showed a significant positive correlation (R = 0.83, *p* < 0.001). Likewise, VL and MD showed a significant positive correlation (R = 0.47, *p* < 0.001). Furthermore, MD and AD showed a significant positive correlation (R = 0.33, *p* < 0.02) (Figure 3, Figure 4 and Figure 5).

### 3.3. Correlation Analysis (Each Material)

The results of the statistical analyses of the various materials are shown in Table 3. A positive correlation was observed for GV, RV and VE in each comparison. However, in PV, only volume–max depth showed a positive correlation, while volume–mean depth and max depth–mean depth showed negative correlations, although these were not significant (*p* > 0.05). For GV and PV, there was a significant positive correlation between volume loss and maximum depth, and no significant correlation between volume loss and mean depth, or between maximum depth and mean depth for PV. For GV and VE, there was a significant positive correlation (r = 0.57 (RV), r = 0.64 (VE)) between maximum depth and mean depth. RV showed no significant correlation (r = 0.26 (max depth), r = 0.44 (mean depth)) between volume loss and either measure of depth, while VE showed a strong positive correlation (r = 0.83) between volume loss and mean depth.

### 3.4. SEM Observation

Representative SEM images of wear facets after localized wear testing are shown in Figure 6, Figure 7, Figure 8 and Figure 9. The lower magnification SEM images show clear differences in wear facet size and worn surface characteristics depending on material and also show clear differences in the form of the wear facets, particularly for GV. The higher magnification SEM images clearly show that there were differences in the size and shape of fillers in the light-cure resin luting cement. SEM images of GV showed quite small (<1 µm) irregular particles. SEM images of PV showed irregular and spherical particles with a broad size range (<1 to 5 µm). SEM images of RV and VE showed irregular particles with a broad size range (<1 to 15 µm).

## 4. Discussion

Our previous study reported that the localized wear of dual-cure resin luting cements had a volume loss of 0.078–0.126 mm^3^ and a maximum depth of 152.2–187.4 µm [16]. Some clinicians like to use dual-cure resin luting cements due to their versatile usage even with their known increase in susceptibility to discoloration. However, in the results of this study, most of the light-cured resin luting cements tested showed wear susceptibility within the range measured for dual-cure cements, and GV was clearly superior in this respect. This suggests that although light-cure resin luting cements may be a better option, when there is no problem with light irradiation of the material, the careful choice of the cement is still important to achieve more favorable results.

Flowable resin-based composites (RBCs) are sometimes used as a substitute for light-cure resin luting cements, because of concerns about the aesthetic quality and wear resistance of cement that is exposed at the margins of a restoration. Our previous study on the wear resistance of flowable RBCs found values for volume loss of 0.025–0.148 mm^3^ and maximum depths of 98.1–210.6 µm [17]. These values are not significantly different from the range found in this study. Our recent study of paste-type RBCs found values for volume loss of 0.034–0.059 mm^3^ and a maximum depth of 110.6–151.5 µm [18]. These values are generally superior to those of light-cured resin luting cements, but in this study GV showed superior wear resistance to that found for paste-type RBCs in the past. These results suggest that wear resistance is not a good reason to use an RBC instead of a resin luting cement, particularly if the RBC must be heated, and that light-cured cements can be used in cases where adequate irradiation can be achieved, such as with ceramic veneers.

On the other hand, the wide range of wear values indicates that there are substantial differences between different materials. This was also observed in this study, in which GV showed superior wear resistance to the other materials. The manufacturer claims that this material uses 150 nm diameter filler particles, which are smaller than those used in other resin luting cements [18]. This claim is supported by the SEM images obtained in this study, which show smaller filler particles for GV than for the other materials investigated. It is possible that the size of the filler particles contributes to the improved wear resistance.

If the size of the filler particles is influencing the wear resistance, it would also be expected to affect the shape of the wear facets, and this can be seen in the SEM images. It is normally assumed that a typical wear facet is roughly hemispherical, with its deepest point at the center and a uniform reduction in depth towards the edges. However, it is clear both from the images and the correlations between the various measurements that things are not so simple. An irregular depression is clearly visible within the wear facet of GV, and that of PV appears to include an elevated region near the center. The fact that the overall correlation between the maximum depth and the mean depth is only 0.33 shows that the facets cannot, in general, be regularly shaped. Further, neither the mean depth nor the maximum depth is correlated with volume loss for all the materials, and some do show a statistically significant correlation for three of the four, in two cases the correlation is with maximum depth, and in one case it is with mean depth. One material (RV) shows no statistically significant correlation between volume loss and either measurement of depth (*p* > 0.05).

This lack of correlation strongly suggests that the wear facets are irregular in shape, and that this irregularity differs from one material to another. As the SEM images show clear differences in the size and shape of the fillers, it is reasonable to hypothesize that these differences are due to filler loss at the surface of the wear facet leading to irregular depressions in the wear surface. This hypothesis is qualitatively consistent with the SEM images of the wear facets, although the irregular depression visible for GV does not appear to have this cause. These results indicate that it is important to use multiple measures of localized wear in order to accurately understand the wear properties of a material, because the correlations between the ways of measuring wear are often weak or non-existent and may vary depending on the material.

In this study, GV showed superior wear resistance to the other materials, and a different wear pattern was observed in the SEM imaging. GV uses smaller filler particles than the other materials, which could plausibly improve wear resistance by reducing the exposure of softer resin matrix at the surface. However, as this study only considered one material with smaller particles, other factors cannot be ruled out.

The main limit of this study is that the cements were exposed to wear in isolation from ceramics and tooth material, a situation that is never found in clinical cases. However, an earlier study compared results for the wear resistance of blocks of resin luting cement and narrow samples between other materials, and found a linear relationship, which suggests that this method is suitable for the comparison of materials [5].

The small number of materials makes it difficult to say anything definite about the causes of the observed differences in wear resistance, but there are few materials of this sort on the market, and so a wide comparison is not possible.

The ideal is for clinicians to choose the best resin luting cement for the situation of each individual patient. However, this requires the clinic to keep a wide range of different materials on hand, and is often impractical. As a result, there is a trend towards the development of universal self-adhesive resin luting cements, which may be used with primer or universal adhesive if necessary to achieve stronger bonding [16]. It is therefore important to also assess the wear resistance of these materials, and compare them to existing dual-cure and light-cure resin luting cements, to determine whether it is appropriate to use them in clinical contexts.

It would also be valuable to perform these tests on experimental versions of these cements, in which the proportion and size of filler or composition of the resin can be varied while keeping other factors constant. This would enable a more definite conclusion on the causes of superior wear resistance, and might contribute to the development of better materials for the future. The results of this study suggest that filler size might be an important factor to vary in such research.

## 5. Conclusions

The light-cure resin luting cements investigated in this study generally show similar wear resistance to dual-cure cements, and also to flowable resin-based composites, but certain cements show superior wear resistance to possible alternative materials, including most paste-type resin-based composites. This class of materials may be suitable for consideration for clinical use from a wear perspective.

## Figures and Tables

**Figure 1 jfb-16-00005-f001:**
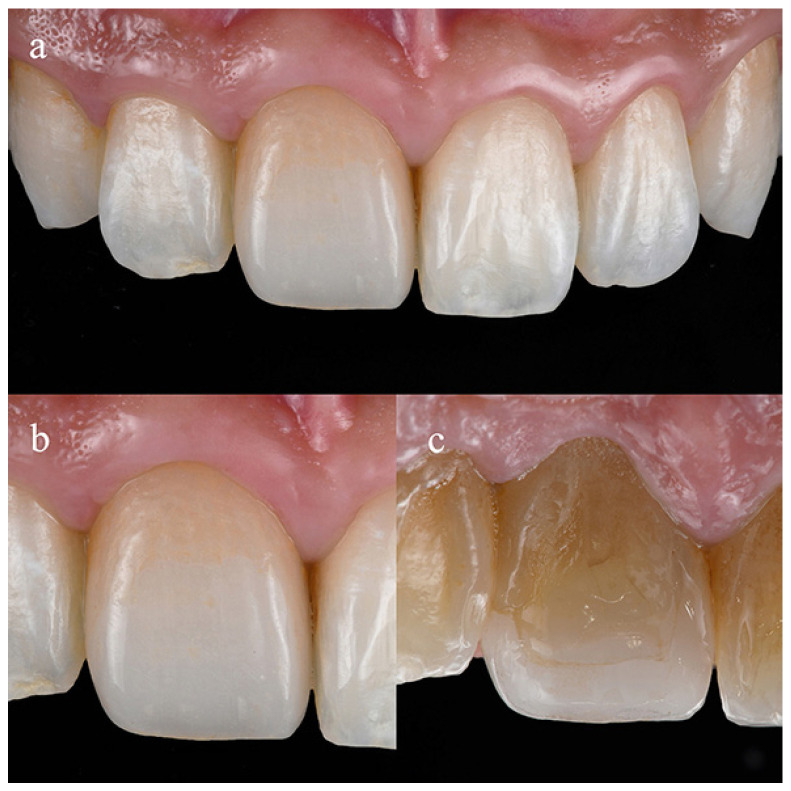
Lithium disilicate press ceramic veneer for tooth #8 after 18 years of usage: (**a**): frontal view; (**b**) enlarged facial view for tooth #8; and (**c**) enlarged lateral view with gap formation in cement line between the veneer and surrounding tooth structure. (Patient is author A.T.).

**Figure 2 jfb-16-00005-f002:**
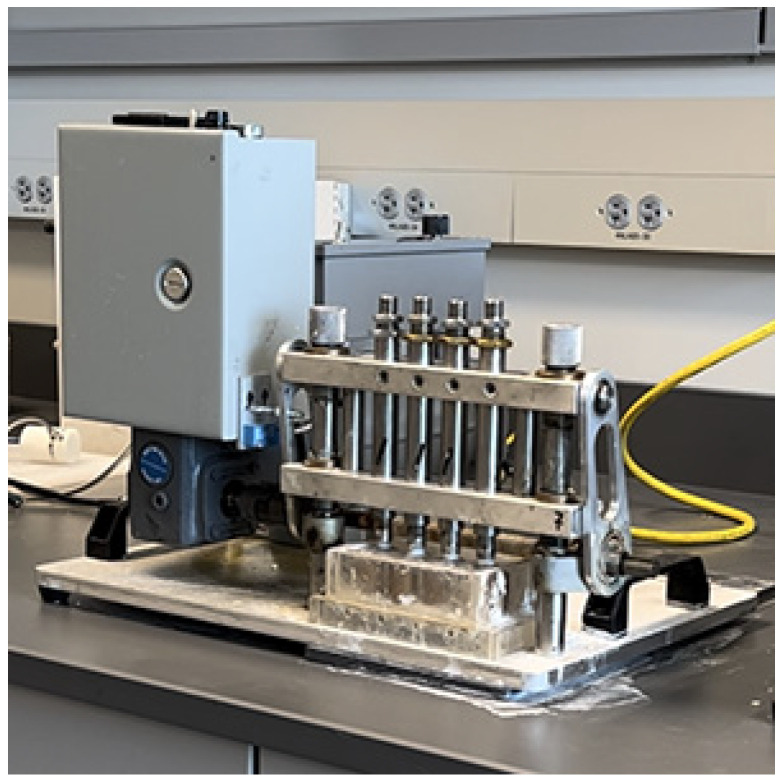
Alabama wear-testing machine in Creighton University School of Dentistry.

**Figure 3 jfb-16-00005-f003:**
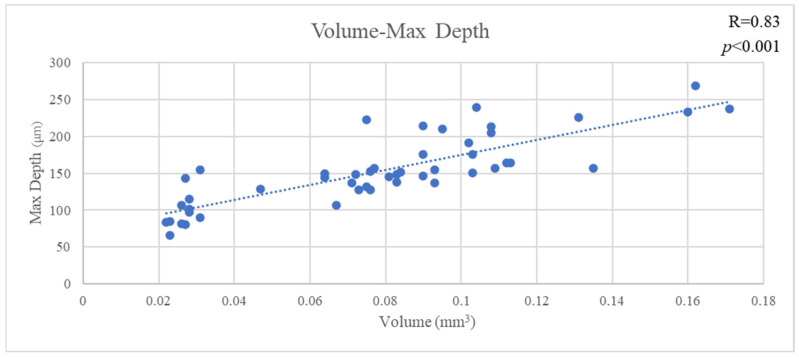
The correlation of volume and max depth.

**Figure 4 jfb-16-00005-f004:**
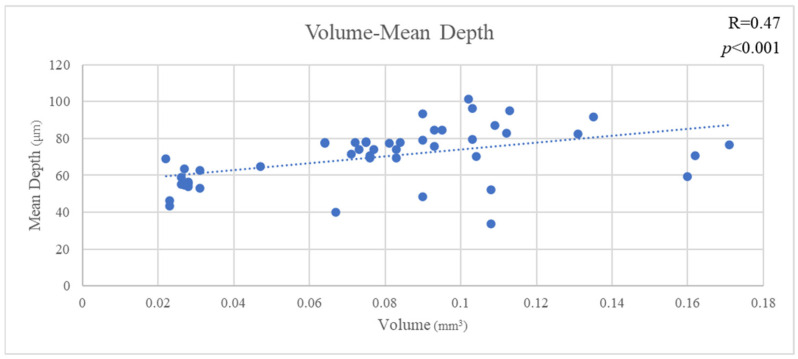
The correlation of volume and mean depth.

**Figure 5 jfb-16-00005-f005:**
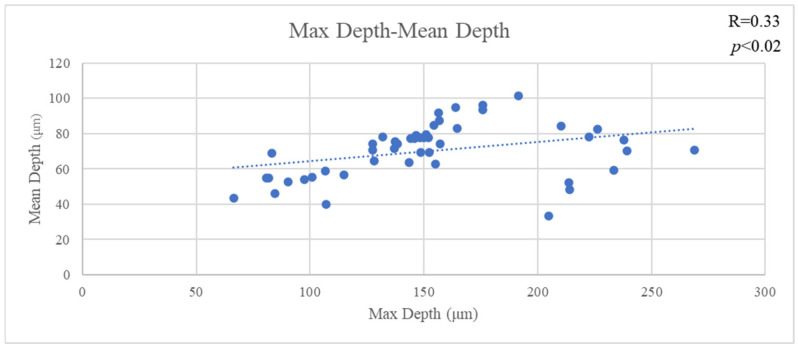
The correlation of max depth and mean depth.

**Figure 6 jfb-16-00005-f006:**
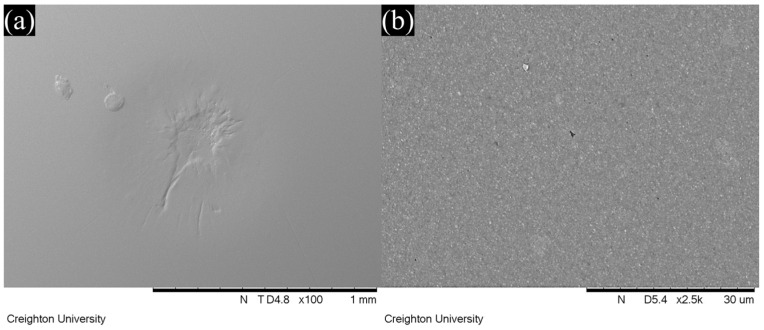
SEM images of the wear facets of G-Cem Veneer as viewed at (**a**) 100× and (**b**) 2500× magnification.

**Figure 7 jfb-16-00005-f007:**
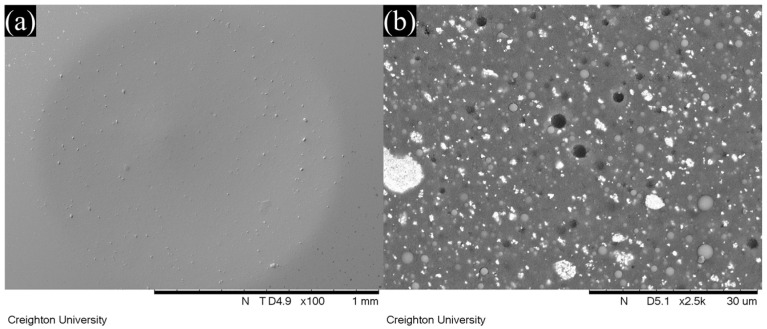
SEM images of the wear facets of Panavia V5 LC as viewed at (**a**) 100× and (**b**) 2500× magnification.

**Figure 8 jfb-16-00005-f008:**
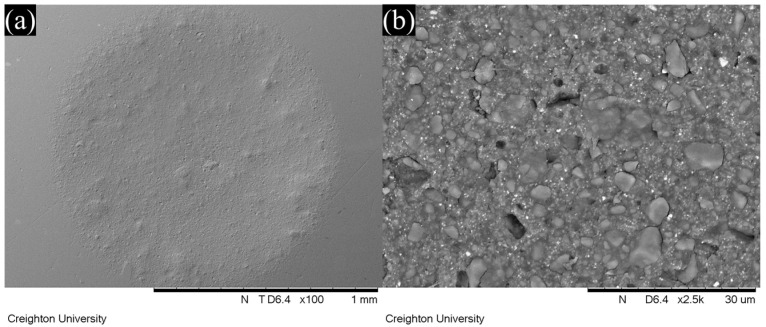
SEM images of the wear facets of RelyX Veneer Cement as viewed at (**a**) 100× and (**b**) 2500× magnification.

**Figure 9 jfb-16-00005-f009:**
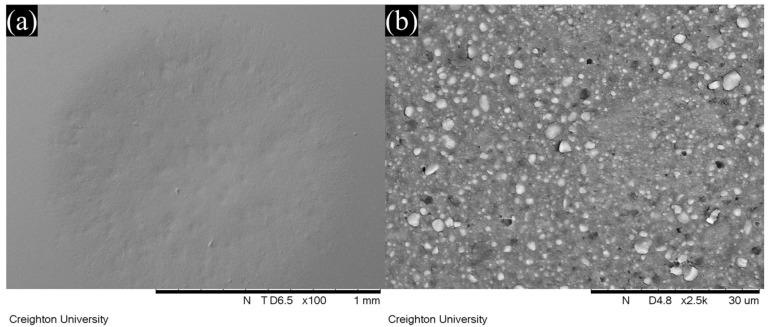
SEM images of the wear facets of Variolink Esthetic LC as viewed at (**a**) 100× and (**b**) 2500× magnification.

**Table 1 jfb-16-00005-t001:** Composition of tested Light-cure Resin Luting Cements.

Materials	Main Components	Manufacture	Study Code
G-Cem Veneer	UDMA, Esterification products of 4,4′-isopropylidenediphenol, ethoxylated and 2-methylprop-2-enoic acid, (octahydro-4,7-methano-1H-indenediyl)bis(methylene) bismethacrylate, 2,2-dimethyl-1,3-propanediyl bismethacrylate, 1,3,5-Triazine-2,4,6-triamine, polymer with formaldehyde, diphenyl(2,4,6-trimethylbenzoyl)phosphine oxide, 2,2′-ethylenedioxydiethyl dimethacrylate, Butylated hydroxytoluene, 2-(2H-benzotriazol-2-yl)-p-cresol, 6-tert-butyl-2,4-xylenol	GC	GV
Panavia Veneer LC	Silanated spherical silica filler, UDMA, Ytterbium trifluorideTriethyleneglycol dimethacrylate, Hydrophilic aliphatic dimethacrylate, Hydrophilic amide monomer, Accelerators, dl-Camphorquinone, Pigments	Kuraray Noritake	PV
RelyX Veneer Cement	Silane Treated Ceramic, BISGMA, TEGDMA, Silane Treated Silica, Reacted Polycaprolactone Polymer, Titanium Dioxide Diphenyliodonium Hexafluorophosphate, N,N-DIMETHYLBENZOCAINE, DL-Camphorquinone	3M ESPE	RV
Variolink Esthetic LC	Matrix: urethane dimethacrylate and further methacrylate monomers.Inorganic fillers: ytterbium trifluoride and spheroid mixed oxide.Additional: initiators, stabilizers and pigments.	Ivolar Vivadent	VE

**Table 2 jfb-16-00005-t002:** Results for localized wear of light-cure resin luting cements.

Materials	Volume Loss (mm^3^)	Maximum Depth (µm)	Mean Depth (µm)
G-Cem Veneer	0.027 (0.003)	100.439 (26.354)	56.053 (7.074) ^a^
Panavia Veneer LC	0.119 (0.030) ^a^	215.958 (27.320)	72.398 (20.853) ^a,b^
RelyX Veneer Cement	0.073 (0.012) ^b^	147.316 (29.896) ^a^	71.900 (12.489) ^a,b,c^
Variolink Esthetic LC	0.096 (0.022) ^a,b^	152.051 (10.910) ^a^	81.531 (7.712) ^b,c^

Same lowercase letter in same vertical column indicates no significant difference.

**Table 3 jfb-16-00005-t003:** Results of Pearson correlation analysis for each material.

Material	Volume Max Depth		Volume Mean Depth		Max Depth Mean Depth	
	R	*p*-Value	R	*p*-Value	R	*p*-Value
GV	0.58	0.05	0.16	0.62	0.57	0.05
PV	0.73	<0.01	−0.10	0.76	−0.35	0.26
RV	0.26	0.41	0.44	0.15	0.54	0.07
VE	0.55	0.06	0.83	<0.01	0.63	<0.01

## Data Availability

The original contributions presented in the study are included in the article, further inquiries can be directed to the corresponding author.

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
