# Peer review of "Wear Resistance of Light-Cure Resin Luting Cements for Ceramic Veneers"

_jfb, 2024, doi:10.3390/jfb16010005_

Round 1

Reviewer 1 Report

Comments and Suggestions for Authors

Dear Authors,

This study addresses an important and interesting topic, covering significant issues that are highly relevant and likely to attract readers in the field of dental materials. However, several limitations in the manuscript warrant attention and improvement. Additionally, numerous grammatical errors require correction by a language expert to enhance the clarity and readability of the study. Below are specific suggestions to improve the quality of your research and ensure it reaches a broader audience:

  • Abstract: The abstract lacks sufficient background on the topic. The resin cements discussed are not novel, and critical details such as the p-value and sample size calculation are missing. The methodology is unclear, and the conclusion requires reformulation to accurately reflect the findings.
  • Keywords: Please revise and organize the keywords using MeSH terms for better indexing.
  • Introduction: There is a long history of the clinical use of ceramic veneers, but the term "combined use" needs clarification. Additionally, the introduction should be revised for grammatical accuracy.
  • Figure 1 Description: The description of long-term veneer usage (16 years) and gap formation requires grammatical revision and better articulation.
  • Heated Composite: Provide a more detailed explanation of heated composite and its role in the study.
  • Dual-Cure Cements: When discussing discoloration, include the role of tertiary amines and mention newer cements that lack this component, which could address discoloration concerns.
  • Resin Cement Classification: Consider starting with a classification of resin cements before delving into their associated challenges, supported by the mentioned figure.
  • Null Hypothesis: Clearly state your null hypothesis.
  • Specimen Preparation: Clarify the number of specimens used and provide the reference for your chosen methodology. Justify why specimens were polished and specify the ISO standard used.
  • AnSur 3D Program: Offer a detailed explanation of the AnSur 3D program utilized in your analysis.
  • SEM Methodology: Provide a comprehensive description of specimen preparation for SEM, including the gold-palladium ratio. The figures are out of focus and require additional explanation to clarify the results. Justify the choice of two magnifications and discuss the "a" and "b" results for each figure.
  • Discussion and Conclusion: Expand the discussion section and provide a detailed analysis of the results. The conclusion should be concise and reformulated to reflect the key takeaways.
  • Limitations and Future Directions: Add a section highlighting the limitations of your study and propose future research directions.
  • References: Ensure the references are accurate and complete.
  • English Revision: The manuscript contains numerous grammatical errors. Professional English editing is essential to improve the quality and clarity of the text.

These revisions are recommended to enhance the scientific rigor, readability, and overall impact of your manuscript.

Comments on the Quality of English Language

The English language should be revised

Author Response

This study addresses an important and interesting topic, covering significant issues that are highly relevant and likely to attract readers in the field of dental materials. However, several limitations in the manuscript warrant attention and improvement. Additionally, numerous grammatical errors require correction by a language expert to enhance the clarity and readability of the study. Below are specific suggestions to improve the quality of your research and ensure it reaches a broader audience:

Abstract: The abstract lacks sufficient background on the topic. The resin cements discussed are not novel, and critical details such as the p-value and sample size calculation are missing. The methodology is unclear, and the conclusion requires reformulation to accurately reflect the findings.

Response: The abstract does cite the p-value, and most of these criticisms are fundamentally inappropriate to an abstract.

Keywords: Please revise and organize the keywords using MeSH terms for better indexing.

Response: We have revised to include a MeSH term. However, it should be noted that MeSH coverage of restorative dentistry is poor.

Introduction: There is a long history of the clinical use of ceramic veneers, but the term "combined use" needs clarification. Additionally, the introduction should be revised for grammatical accuracy.

Response: We have deleted the term “combined”. Additionally, the introduction has been checked by a professional native-speaking editor and is grammatically accurate.

Figure 1 Description: The description of long-term veneer usage (16 years) and gap formation requires grammatical revision and better articulation.

Response: There were no significant problems with the grammar of this paragraph. However, we have revised it slightly to make the point clearer. If the comment actually refers to the figure legend: it is a figure legend, and the grammatical standards for body text do not apply.

Heated Composite: Provide a more detailed explanation of heated composite and its role in the study.

Response: Heated composite plays no role in the study. It is a material used by some clinicians out of concern for the wear resistance of the cements. We do not recommend it, and raise problems for it in the text already.

Dual-Cure Cements: When discussing discoloration, include the role of tertiary amines and mention newer cements that lack this component, which could address discoloration concerns.

Response: This is already mentioned in the manuscript. It is only a single sentence, because it is not of major importance.

Resin Cement Classification: Consider starting with a classification of resin cements before delving into their associated challenges, supported by the mentioned figure.

Response: There is no mentioned figure, we do discuss different types of resin cements, and we do not see the relevance of this comment to the draft.

Null Hypothesis: Clearly state your null hypothesis.

Response: Not every study uses a null hypothesis, even when comparing some statistical measures. This study does not, because there is no sensible null hypothesis. We are not interested in simply establishing that there is a difference between some of the measures between the materials. We are interested in how these differences relate to the nature of the materials.

Specimen Preparation: Clarify the number of specimens used and provide the reference for your chosen methodology. Justify why specimens were polished and specify the ISO standard used.

Response: The number of specimens is discussed in the first paragraph of section 2.2. The references are provided in section 2.3. It appears that two placeholders for this information survived to the final draft, for which we apologise, but the information was present in the manuscript as submitted, and a human reviewer should have noticed that. Polishing was conducted to approximate clinical conditions, and we have added a note to that effect, but there is no relevant ISO standard.

AnSur 3D Program: Offer a detailed explanation of the AnSur 3D program utilized in your analysis.

Response: This is standard software for this analysis, and none of us are experts in computer software. We do not believe that this is necessary.

SEM Methodology: Provide a comprehensive description of specimen preparation for SEM, including the gold-palladium ratio. The figures are out of focus and require additional explanation to clarify the results. Justify the choice of two magnifications and discuss the "a" and "b" results for each figure.

Response: We have added the gold-palladium ratio. It is the standard for the apparatus. The specimens were coated and magnified as they were, so the description is already comprehensive. The figures are not out of focus, and there is extensive discussion of them in the Discussion. The two magnifications were chosen to show the overall shape of the wear facet and the detailed

Reviewer 2 Report

Comments and Suggestions for Authors

1. why did authors use ceramic veneers instead of composite veneers.

2. add some dental composite's history in the introduction section and add all recommended article into it.

A. Tribological behavior of dental resin composites: A comprehensive review

B. Ranking Analysis of Tribological, Mechanical, and Thermal Properties of Nano Hydroxyapatite Filled Dental Restorative Composite Materials Using the R-Method

3. Section 2.3:

A. add line diagram of wear machine

B. author mentioned "78.5 N". did author vary the load or not. if yes. explain what is the minimum load.

C. why did author use  unplasticized polymethyl methacrylate as abrasive media.

4. add limitation of current work.

5. add a graph of wear loss with load or cycle.

Author Response

Comments and Suggestions for Authors

    why did authors use ceramic veneers instead of composite veneers.

Response: We did not use ceramic veneers. The testing was of the cements alone, and not of the veneers. The cements could, in principle, be used with composite veneers, and the results would also apply in such cases. The introduction discusses the reasons why the contrast with ceramic veneers is particularly important, at considerable length.

    add some dental composite's history in the introduction section and add all recommended article into it.
    Tribological behavior of dental resin composites: A comprehensive review
    Ranking Analysis of Tribological, Mechanical, and Thermal Properties of Nano Hydroxyapatite Filled Dental Restorative Composite Materials Using the R-Method

Response: We have added a brief reference to the first of these papers, but the second is not germane to our research.

    Section 2.3:
    add line diagram of wear machine

Response: We have published such a diagram in several previous papers, and have added the diagram to this manuscript as well. Note that this diagram is reused, because the apparatus is the same.

    author mentioned "78.5 N". did author vary the load or not. if yes. explain what is the minimum load.

Response: As described in the paper, the load varies over the cycle. The antagonist is removed from the material entirely at one end, so the minimum load is zero. We have clarified this in the paper.

    why did author use  unplasticized polymethyl methacrylate as abrasive media.

Response: This is the standard medium for Alabama wear tests. We are following the standard to ensure comparability of results.

    add limitation of current work.

Response: This is covered in the last four paragraphs of the Discussion, along with suggestions for further work.

    add a graph of wear loss with load or cycle.

Response: As clearly described in the paper, we measured wear by comparing scans of the surface before and after the wear simulation. As a result, we have one data point per specimen, and no data for different loads or cycle numbers, and therefore it is not possible to plot these graphs.

Round 2

Reviewer 1 Report

Comments and Suggestions for Authors

The authors did not consider the comments.

Comments on the Quality of English Language

Major

Reviewer 2 Report

Comments and Suggestions for Authors

No further comments.